# Exposed Phosphatidylserine as a Biomarker for Clear Identification of Breast Cancer Brain Metastases in Mouse Models

**DOI:** 10.3390/cancers16173088

**Published:** 2024-09-05

**Authors:** Lulu Wang, Alan H. Zhao, Chad A. Arledge, Fei Xing, Michael D. Chan, Rolf A. Brekken, Amyn A. Habib, Dawen Zhao

**Affiliations:** 1Department of Biomedical Engineering, Wake Forest University School of Medicine, Winston-Salem, NC 27157, USA; lulwang@wakehealth.edu (L.W.);; 2School of Medicine, University of North Carolina, Chapel Hill, NC 27599, USA; alan_zhao@med.unc.edu; 3Department of Cancer Biology, Wake Forest University School of Medicine, Winston-Salem, NC 27157, USA; fxing@wakehealth.edu; 4Department of Radiation Oncology, Wake Forest University School of Medicine, Winston-Salem, NC 27157, USA; mchan@wakehealth.edu; 5Hamon Center for Therapeutic Oncology Research, Department of Surgery, UT Southwestern Medical Center, Dallas, TX 75390, USA; rolf.brekken@utsouthwestern.edu; 6Department of Neurology, UT Southwestern Medical Center, Dallas, TX 75390, USA; amyn.habib@utsouthwestern.edu; 7Department of Translational Neuroscience, Wake Forest University School of Medicine, Winston-Salem, NC 27157, USA

**Keywords:** Phosphatidylserine (PS), brain micrometastases, vascular endothelial cells, blood–tumor barrier (BTB), imaging

## Abstract

**Simple Summary:**

The prognosis of brain metastasis is extremely poor, partly because of the concurrence of multiple brain lesions and their limited access to current systemic therapies. Applying a phosphatidylserine (PS)-targeting antibody, we identified abundant PS on the vasculature of brain metastases in mouse models. Given its location on the luminal surface of tumor blood vessels, exposed PS appears to be an ideal target for both diagnostic and therapeutic agents, which otherwise have difficulty penetrating the blood–tumor barrier (BTB) of brain metastases.

**Abstract:**

Brain metastasis is the most common intracranial malignancy in adults. The prognosis is extremely poor, partly because most patients have more than one brain lesion, and the currently available therapies are nonspecific or inaccessible to those occult metastases due to an impermeable blood–tumor barrier (BTB). Phosphatidylserine (PS) is externalized on the surface of viable endothelial cells (ECs) in tumor blood vessels. In this study, we have applied a PS-targeting antibody to assess brain metastases in mouse models. Fluorescence microscopic imaging revealed that extensive PS exposure was found exclusively on vascular ECs of brain metastases. The highly sensitive and specific binding of the PS antibody enables individual metastases, even micrometastases containing an intact BTB, to be clearly delineated. Furthermore, the conjugation of the PS antibody with a fluorescence dye, IRDye 800CW, or a radioisotope, ^125^I, allowed the clear visualization of individual brain metastases by optical imaging and autoradiography, respectively. In conclusion, we demonstrated a novel strategy for targeting brain metastases based on our finding that abundant PS exposure occurs on blood vessels of brain metastases but not on normal brain, which may be useful for the development of imaging and targeted therapeutics for brain metastases.

## 1. Introduction

Brain metastasis is the most frequently occurring intracranial malignancy in adults [1,2]. The incidence of brain metastasis seems to have increased and may be the paradoxical result of effective therapies against primary cancer. In part, this is due to the fact that anti-cancer drugs that show efficacy against visceral cancers have poor penetration of the blood–tumor barrier (BTB), which means that brain metastases are significantly undertreated [3,4,5,6]. Many brain metastasis patients exhibit multiple tumors at the time of diagnosis. Even in the event of a solitary metastasis, it is believed that the brain may be seeded with many radiographically invisible metastases [7,8]. Whole-brain radiation therapy (WBRT) alone or in combination with surgical resection or stereotactic radiosurgery is thus the standard of care for brain metastasis patients. However, WBRT is often associated with neurological complications that preclude sufficient radiation doses to effectively treat the lesions. The prognosis of brain metastasis is extremely poor, with a median survival of 8–16 months, even with combinatorial treatment [9,10]. Thus, there is an urgent need to develop new diagnostic and therapeutic agents that are specific and effective for brain metastases.

Multiple strategies have been exploited to improve access of systemic diagnostic and therapeutic agents to brain metastases [11,12]. However, limited success has been reported with the physical or chemical approaches to open BTB, primarily owing to the inhomogeneous disruption of BTB and, more importantly, unwanted damage of the blood–brain barrier (BBB) in the normal brain, resulting in neurological toxicity from drugs or blood components entering the normal brain [6]. Taking advantage of intravascular receptors that mediate transcytosis across the BBB, several ligands or antibodies that specifically bind to such receptors have been conjugated directly or indirectly with therapeutic compounds [13]. Despite leading to increased drug concentrations in brain tumors, this strategy also causes toxicity to the normal brain because receptors such as transferrin, insulin, and low-density lipoprotein are widespread in the normal brain [14]. Clearly, the discovery of a biomarker that is specific to brain metastasis will be critical for the development of brain metastasis-targeted diagnosis and therapy. Unlike visceral cancers, in which tumor capillaries are largely leaky due to extensive angiogenesis, the clinically occult or micro-metastases are impermeable to diagnostic or therapeutic agents. Ideally, this biomarker needs to be accessible to systemically administered diagnostic and therapeutic agents.

Phosphatidylserine (PS), the most abundant anionic phospholipid of the cell membrane, is normally constrained to the inner leaflet of the plasma membrane of healthy cells. The asymmetric PS distribution is maintained by flippase or aminophospholipid translocase, which translocates PS from the outer to the inner leaflet of the lipid bilayer membrane [15,16]. PS becomes externalized on the outer leaflet of the plasma membrane of cells when the flippase becomes inhibited or PS transporters such as scramblase become activated by calcium (Ca^2+^) fluxes [17]. Loss of PS asymmetry is observed during programmed cell death—apoptosis. However, PS exposure can occur in the endothelial cells of tumor blood vessels and cancer cells [18,19]. These cells are viable, and PS exposure is inducible and reversible, which is distinct from the irreversible process of PS externalization during apoptosis [15,20,21]. The exact mechanisms of this phenomenon remain unknown. However, oxidative stresses in the tumor microenvironment (TME) are believed to plausibly contribute to PS exposure, e.g., reactive oxygen species (ROS), acidic and hypoxic tumors, and chronic inflammation. We and others have previously reported various levels of PS-positive tumor blood vessels, ranging from 5% to 40% among multiple tumor models [22,23]. However, there is no prior research that has investigated the extent of PS exposure in brain metastasis.

In the present study, we applied a monoclonal PS-targeting antibody, 1N11, to assess PS exposure on the vasculature of brain metastases in mouse models. After systemic administration of 1N11, PS exposure on tumor vascular ECs was quantified by detecting 1N11+ CD31+ ECs with fluorescence microscopy imaging. Notably, the 1N11+ fluorescence signal was only seen in brain metastases, coinciding well with CD31+ tumor vascular ECs. Examination of a series of whole-mount brain sections revealed that 1N11+ fluorescence signals clearly demarcated individual brain metastases. Since previous works suggest that PS exposure in non-apoptotic cells is possibly caused by oxidative stresses in the TME, we investigated the plausible contribution of oxidative stress-related factors, including tumor hypoxia and inflammatory cytokines such as tumor necrosis factor alpha (TNF-α). To investigate the potential of targeting exposed PS for sensitive and specific imaging of brain metastases, we labeled 1N11 with a fluorescence dye, IRDye 800CW, or a radiotracer, I-125, to perform optical imaging and autoradiographic imaging of brain metastases in mice. Our results showed that PS was abundantly exposed on the blood vessels of brain metastases, while there was no PS exposure detected in normal brain. Moreover, the vascular luminal surface-exposed PS might serve as a biomarker particularly useful for developing brain metastasis-targeting agents, which otherwise have difficulty passing through the BTB to brain metastases.

## 2. Materials and Methods

### 2.1. Reagents and Cell Lines

The human monoclonal antibody, 1N11, binds to PS, while Aurexis, a human monoclonal antibody, binds to an irrelevant antigen (*S. aureus* clumping factor A), serving as a negative control antibody. Both antibodies were generously provided by Drs. Philip Thorpe and Rolf Brekken (UT Southwestern Medical Center at Dallas). Triple-negative breast cancer is a highly aggressive breast cancer subtype that tends to metastasize to the brain. Both 4T1 and 231-Br cells were derived from triple-negative breast cancer. Murine breast cancer 4T1 cells and human umbilical vein endothelial cells (HUVECs) were purchased from American Type Culture Collection (ATCC, Manassas, VA, USA). The brain-tropic human breast cancer MDA-MB-231/BR-GFP (231-Br) cell line was a kind gift from Dr. Patricia Steeg (NIH/NCI).

### 2.2. Breast Cancer Brain Metastasis Models

All animal procedures were approved by the Institutional Animal Care and Use Committee of Wake Forest University School of Medicine and the University of Texas Southwestern Medical Center. Murine breast cancer 4T1 cells and brain-tropic human breast cancer 231-Br cells were incubated in Dulbecco’s modified Eagle’s medium (DMEM) with 10% FBS, 1% L-Glutamine and 1% penicillin–streptomycin at 37 °C with 5% CO_2_. Once 80% confluence was reached, the cells were harvested and suspended in serum-free medium. Immunocompetent female BALB/c mice (*n* = 6; 6–8 weeks old; NCI, Frederick, MD, USA) were used for the 4T1 model, while female nude mice (*n* = 16; BALB/c nu/nu, 6–8 weeks old; NCI or The Jackson Laboratory) were used for the human breast cancer 231-Br model. The mice were anesthetized with the inhalation of 2% isoflurane. Then, 2 × 10^5^ 4T1 cells or 231-BR cells (in 100 μL of serum-free medium) were injected directly into the left ventricle of a mouse heart under the imaging guidance of a small animal ultrasound (Vevo 770, VisualSonics; Toronto, ON, Canada).

### 2.3. Longitudinal MRI Monitoring of Development of Brain Metastases

MRI was initiated two weeks after tumor implantation and repeated once a week for up to three weeks. Animals were sedated with 3% isoflurane and maintained under general anesthesia (1.5% isoflurane). Animal body temperature and respiration were monitored and maintained constant throughout the experiment. MR measurements were performed using a 9.4T horizontal bore magnet with a Varian INOVA Unity system (Palo Alto, CA, USA) or a 7T Bruker BioSpec 70/30 USR scanner (Bruker Biospin, Rheinstetten, Germany). A tail vein of a mouse was catheterized using a 27G butterfly for Gd-DTPA (Magnevist^®^; Bayer HealthCare, Wayne, NJ, USA) contrast agent administration. High-resolution multi-slice (14 slices with 1 mm thick) T_2_-weighted coronal images, covering from the frontal lobe to the posterior fossa, were acquired with a fast spin echo multi-slice sequence (TR/TE = 2500 ms/48 ms, 8 echo trains, matrix: 256 × 256, FOV 20 × 20 mm, resolution: 78 × 78 µm^2^ in plane). T_1_-weighted contrast-enhanced images were acquired with a spin echo multi-slice sequence (TR/TE = 400 ms/20 ms, matrix: 256 × 256, FOV 20 × 20 mm). We determined tumor volume on T_2_-weighted images by manually outlining the enhancing portion of the mass on each image by using standard ‘‘browser’’ software provided with the Varian Inova imaging system or the Bruker system.

### 2.4. Detection and Quantification of Exposed PS In Vivo

Immediately after the last MRI follow-up of the 231-Br mice (*n* = 6), 150 μg of 1N11 or the control body Aurexis were injected i.v. and allowed to circulate for 4 h. The mice were anesthetized, exsanguinated, and perfused with heparinized saline. The mouse brains were dissected and frozen for the preparation of cryosections. We analyzed the entire brain of each animal by immunohistochemistry. The vascular endothelium was stained using a rat anti-mouse CD31 antibody (1:20; Serotec Inc., Raleigh, NC, USA) followed by Alexa Fluor 350-labeled goat anti-rat IgG (1: 200; BD Biosciences, San Jose, CA, USA). 1N11 or Aurexis was detected using goat anti-human IgG conjugated to Cy3 (1:500; Jackson Immunoresearch Laboratory, West Grove, PA, USA). The immunostained sections of the entire brain were evaluated under Zeiss LSM510 (Carl Zeiss MicroImaging, Inc., Thornwood, NY, USA), and the individual 10× fluorescence images were automatically stitched by using SoftWoRx 4.2.1 (Applied Precision, Issaquah, WA, USA). Doubly labeled endothelial cells (i.e., CD31-positive/1N11-positive) were identified on merged images. The percentage of doubly positive vessels was calculated as follows: (mean number of yellow vessels per field/mean number of total vessels) × 100. Ten random fields containing metastases were evaluated for each section.

### 2.5. Immunohistochemical Detection of Hypoxia and TNF-α in Brain Metastasis

To detect hypoxia, pimonidazole (60 mg/kg, Hypoxyprobe, Burlington, MA, USA) was injected intravenously 60 min before sacrifice. Pimonidazole was recognized by primary Mab1 (1:100; Hypoxyprobe) and followed by rabbit anti-mouse secondary conjugated to AMCA (1:300; Jackson Immunoresearch Laboratory). For TNF-α staining, anti-TNF-α (1:200; R&D Systems, Minneapolis, MN, USA) was followed by horseradish peroxidase (HRP)-conjugated goat anti-rat secondary antibody (1:1000; Santa Cruz Biotechnology, Dallas, TX, USA). After a PBST wash, sections were immersed in DAB (3,3′-diaminodbenzidine; Thermo Fisher Scientific, Waltham, MA, USA) and then observed under the microscope.

### 2.6. Near-Infrared Fluorescence Imaging

1N11 and Aurexis F(ab’)_2_ fragments were generated by reacting antibodies with pepsin at a molar ratio of 1:130 (antibody–pepsin) for 1 h at 37 °C. F(ab’)_2_ fragments (MW = 110 kDa) were purified by FPLC using an S-200 column (Pharmacia, Piscataway, NJ, USA) and PBS running buffer. F(ab’)_2_ was then reacted with an N-hydroxysuccinimide ester derivative of IRDye 800CW (Li-COR, Lincoln, NE, USA) at a molar ratio of 1:10 (F(ab’)_2_:dye) for 2 h at room temperature. Unreacted dye was separated from the conjugate using a PD-10 desalting column (GE Healthcare, Uppsala, Sweden). Analyses of the final product, based on the absorbance of the dye at 778 nm and the absorbance of the F(ab’)_2_ at 280 nm, showed that it consisted of approximately 2 molecules of dye bound to each F(ab’)_2_ fragment. The products are referred to as 800CW-1N11 or 800CW-Aurexis throughout this manuscript. Immediately after the last MRI follow-up at 5 weeks post-intracardiac injection, 800CW-1N11 or 800CW-Aurexis (2 nmol/mouse) was injected into a tail vein of the brain metastasis mice. Twenty-four hours later, the mice were sacrificed and perfused, and their brains were dissected. Ex vivo fluorescence imaging (excitation, 671–705 nm; emission, 730–950 nm) was performed using a Maestro imaging system (CRI Inc., Woburn, MA, USA). Brain tissues were then embedded in optimal cutting temperature compound (O.C.T.) and then transferred to a −80 °C freezer. On the second day, a series of coronal sections (10 µm) of the entire brain was cut with a cryostat. The sections containing tumor tissues were identified by H&E staining. The adjacent frozen sections were then used for ex vivo fluorescence imaging. The ex vivo images of the thin frozen sections were acquired by the same Maestro system. Typically, an exposure time of 2.0 s was applied for image acquisition. Fluorescence images were processed with the Maestro software 2.8. Lastly, the frozen sections were then stained with anti-CD31 and correlated with the 800CW-1N11 signals using an Envos microscope (AMG, Bothell, WA, USA) equipped with near-infrared (NIR) filters.

### 2.7. Autoradiographic Imaging of I-125-Labeled 1N11

Iodine-125 (^125^I) was purchased from Perkin Elmer (Waltham, MA, USA). Pre-coated iodination tubes and Protein A agarose were from Pierce Biotechnology (Rockford, IL, USA). Instant thin-layer chromatography plates (ITLC-SG) were from Pall Life Sciences (East Hills, NY, USA). F(ab’)_2_ fragments were generated by reacting antibodies with pepsin at a molar ratio of 1:130 (antibody–pepsin) for 1 h at 37 °C. F(ab’)_2_ fragments were purified on an FPLC S-200 column (Pharmacia, Piscataway, NJ, USA). The F(ab’)_2_ fragments were then radioiodinated using the indirect IODO-GEN method (Pierce Biotechnology). Briefly, 1–3 mCi of iodine was activated in 100 µL iodination buffer (125 mM Tris-HCL, pH 6.8, 150 mM NaCl) in a pre-coated iodination tube and then reacted with 0.2–0.6 mg F(ab’)_2_ in 100 µL iodination buffer in a separate uncoated tube. Free iodine was removed with Bio-Spin 6 gel filtration columns (Bio-Rad Laboratories, Hercules, CA, USA) that were pre-blocked with iodination buffer containing 10% FBS. Radio-TLC analysis was used to determine iodination efficiency on a Rita Star Radioisotope TLC Analyzer (Straubenhardt, Germany) using ITLC-SG plates. Immediately after an anatomic MRI, each mouse was given i.v. 60 µCi ^125^I-1N11 F(ab’)_2_. Forty-eight hours later, the mice were perfused, and mouse brains were dissected. Using a mouse brain matrix, 4 mm thick brain-bearing brain metastases correlating with the MRI were cut from each mouse brain and laid on an autoradiographic film. After a 12 h incubation, the film was read using Cyclone Plus Storage Phosphor (PerkinElmer, Waltham, MA, USA). The spatial localizations of ‘hot spots’ on radiographic images were correlated with MRI stacked images. The signal intensity of the tumor versus the normal brain was also calculated.

### 2.8. Statistical Analysis

Statistical analysis was performed using Microsoft 365 Excel. Data were presented as mean ± SD. Statistical significance was determined by the Student’s *t*-test. All *t*-tests were two-tailed, unpaired, and considered statistically significant if *p* < 0.05.

## 3. Results

### 3.1. MRI Detects Brain Metastases and Evaluates the BTB Permeability

Longitudinal MRI was performed to detect and follow up on the intracranial development of brain metastases in mice. As shown in Figure 1, multi-focal brain metastases started to appear on T_2_-weighted images 3 weeks after the intracardiac injection of 231-Br cells, many of which became apparently larger on the follow-up MRI at week 4. T_1_-w contrast-enhanced images at the initial scan revealed a small fraction of the lesions with high signal intensity, although the number of the enhanced lesions increased at the follow-up scan, which indicated that the BTB in many brain metastases was intact (Figure 1a). For a total of 464 brain metastases that were identified on T_2_-w images, only 160 of them were found to be permeable to the MRI contrast agent, Gd-DTPA (Figure 1b). Interestingly, our data showed that there was no significant correlation between tumor volume and BTB permeability (Figure 1b). In good agreement with previous studies by others and us [6,24,25], these data clearly showed the heterogeneous BTB permeability among individual brain metastases and the evolution of the BTB disruption with the intracranial growth of the metastases.

### 3.2. PS Externalization in Vascular ECs of Brain Metastases but Not Normal Brain

The 231-Br model was used in this study, and MRI scans were performed to confirm the intracranial development of multiple brain metastases and heterogeneous permeability in the BTB of individual brain metastases (Figure 2a). Immediately after MRI, 1N11 or Aurexis (150 μg) were injected intravenously into 231-Br mice. Four hours later, the mice were exsanguinated and perfused to wash out unbound antibodies in the circulation. H&E-stained brain sections correlated well with MRI but apparently depicted additional micrometastases that were undetectable by MRI (Figure 2b,c). Immunohistochemical staining of 1N11 was then performed on frozen sections of the entire mouse brain, which were also co-stained with anti-CD31 antibody for vascular ECs (Figure 2d–j). As shown in Figure 2d, fluorescence microscopic imaging revealed extensive 1N11-positive regions on a whole-brain section, which coincided with individual metastases identified on the H&E staining (Figure 2c), including micrometastases (Figure 2d). The superior fluorescence contrast demarcated every tumor lesion from the surrounding normal brain (Figure 2c,d). Higher magnification images further showed that 1N11 co-localized with almost every CD31-positive tumor vessel (Figure 2g–j and Figure 3a–c). Blood vessels in the nearby normal brain were not stained by 1N11 (Figure 2i,j and Figure 3c). In contrast, there was no positive Aurexis signal in brain metastases (Figure 3d–f). Thus, staining with 1N11 was antigen-specific. Quantitative analysis determined that 93% of blood vessels of brain metastases were found to be PS-positive (Figure 3g). Intriguingly, positive PS staining was not observed in the extravascular space (Figure 2i and Figure 3c), indicating that biological macromolecules, such as proteins, were unable to bypass the BTB to reach the tumor parenchyma of the 231-Br brain metastases, even though some lesions were permeable to the small-molecule MRI contrast agent, Gd-DTPA. Together, our data demonstrate that extensive PS exposure occurs in the vascular endothelial cells of brain metastases but not the normal brain, and positive PS staining enables the clear delineation of individual brain metastases, even micrometastases, from the normal brain.

### 3.3. Inflammatory Cytokines Are Likely Responsible for PS Exposure in Brain Metastases

It is believed that oxidative stresses, such as tumor hypoxia, are important in the induction of PS exposure [18,26]. We applied the hypoxic marker pimonidazole to study tumor hypoxia in brain metastases. However, there was no positive pimonidazole staining in any of the brain metastases (Figure 4a). The lack of tumor hypoxia in brain metastases was also reported by Lorger and Felding-Habermann, who studied several breast cancer brain metastasis mouse models, including the 231-Br model [27]. These data suggest that brain metastases at an early stage of development are supplied with adequate oxygen from blood vessels; thus, tumor hypoxia is an unlikely contributor to PS exposure in these small brain metastases. Recent studies have reported that inflammatory cytokines, such as TNF-α, are upregulated in brain metastasis and highly associated with the extravasation of cancer cells from blood vessels and their colonization in the brain parenchyma [28]. TNF-α signaling is known to generate reactive oxygen species (ROS) [29], and ROS can induce lipid peroxidation in the cell membrane and, thus, changes in membrane properties [30]. Thus, we investigated if TNF-α may induce PS exposure. We treated HUVECs with recombinant TNF-α and stained PS with 1N11. Indeed, abundant PS was detected on the surface of the cells treated with TNF-α but absent in the non-treated cells (Figure 4b). We further studied TNF-α expression in brain metastases. The overexpression of TNF-α was consistently evidenced in brain metastases by immunohistochemistry (Figure 4c). While such inflammatory cytokines as TNF-α may be linked to PS exposure, we cannot rule out plausible contributions from other stress-relevant factors.

### 3.4. Targeting PS Enables Sensitive and Specific Imaging of Brain Metastases

To explore the possibility of targeting PS for brain metastases imaging, we created the F(ab’)_2_ fragment of 1N11 from the full-length 1N11 and then conjugated the 1N11-F(ab’)_2_ with an NIR dye, IRDye800CW, to form 800CW-1N11. We applied NIR optical imaging to image the ability of 800CW-1N11 to target exposed PS in brain metastases. In addition to the 231-Br model, we established a brain metastasis model by intracardiacally injecting the murine breast cancer 4T1 cells. Unlike the 231-Br model bearing multifocal brain metastases, the 4T1 model was found to generally form fewer lesions, often a solitary tumor (Figure 5). At 24 h after i.v. 800CW-1N11, while NIR imaging of the whole brains revealed multiple regions with high NIR signals in the 231-Br model, a single distinct signal was observed in the 4T1 brain metastasis (Figure 5a). To interrogate the brain metastasis-targeting sensitivity and specificity of the 800CW-1N11 imaging, we took a more in-depth look at the signals of 800CW-1N11 throughout the brain sections bearing 4T1 brain metastasis. As shown in Figure 5b, a brain lesion appeared on multiple MRI slices, and the corresponding H&E sections confirmed the brain metastasis. Ultrathin unstained cryosections (10 μm) adjacent to the H&E sections were then imaged with NIR imaging. Notably, the improved contrast of the tumor with distinct tumor margins was detected in each tumor-bearing whole-brain section by NIR imaging, which correlated well with histological staining (Figure 5b). The ratio of the mean signal intensity of tumor to contralateral normal brain, obtained from the ultrathin sections, was 5.8, which was significantly higher than that of the control antibody conjugates, 800CW-Aurexis (1.6; *p* < 0.05; Figure 5c). Like the 1N11 antibody, the majority of 800CW-1N11 was found to bind to tumor vascular endothelial cells despite some extravascular signals observed under the fluorescence microscope (Figure 5d–f). The extravascular signals likely resulted from the leakage of 800CW-1N11 via the disrupted BTB, which was evidenced on T_1_-w contrast images (Figure 5b).

To further exploit the potential of PS-targeted multimodal imaging, we radiolabeled 1N11 F(ab’)_2_ with iodine-125, ^125^I-1N11, and injected the conjugates (60 µCi) into a tail vein of the mice bearing 231-Br brain metastases or the healthy mice (Figure 6a). Forty-eight hours later, the mice were perfused, and mouse brains were dissected. Autoradiography was conducted on 4 mm thick brain sections that aligned with the MRI. To enable spatial correlation of the hot spots on autoradiographs (4 mm thick slice) with tumor lesions on MRI (1 mm thick slice), four consecutive 1 mm MRI images were stacked to generate a synthetic MR image that contained all the tumor lesions from each section (Figure 6a). As shown in Figure 6b, the autoradiograph captured multiple hot spots on the tumor-containing brain. Clearly, there was a good spatial correlation between the hot spots on the autoradiograph and tumor lesions on the MRI (Figure 6a,b; Appendix A). In contrast, a clean background devoid of signal was observed on the normal brain (Figure 6b). The uptake of ^125^I-1N11 was significantly higher in the individual lesions than in the normal brain, with a ratio of 3.6 ± 0.8 (*p* < 0.05; Figure 6c). Taken together, these data demonstrate the high sensitivity and specificity of the PS-targeted imaging of brain metastases in mouse models.

## 4. Discussion

A significant finding from this study is that abundant PS is externalized on the luminal surface of blood vessels of brain metastases, which can be recognized by the systemically administered PS-targeting antibody. While previous studies reported various levels of PS-positive blood vessels among cancer types [22], our data detected PS expression in ~90% of the vascular ECs of brain metastases (Figure 2 and Figure 3). Importantly, PS exposure was found to be specific to brain metastases since there was no PS detected in normal brain, and the control antibody, Aurexis, showed no positive staining in the blood vessels of brain metastases (Figure 3). Unlike other tumor models, in which significant angiogenesis commonly leads to enhanced vascular permeability, the intracardiac 231-Br model contains many tumor lesions with an intact BTB, which was found impermeable to the small-molecule MRI contrast agent, Gd-DTPA (Figure 1 and Figure 2). Along with micrometastases detected by histology, these small lesions may be comparable to those occult brain lesions in humans. Notably, positive PS staining co-localized completely with CD31+ tumor vascular ECs. Neither the PS antibody nor the control antibody, Aurexis, was seen in extravascular tumor tissues, even in those brain metastases that were obviously enhanced by Gd-DTPA (Figure 2). These data clearly indicated that biological macromolecules, such as proteins, were largely inaccessible to the tumor parenchyma of the 231-Br brain metastases. These results are in good agreement with previous studies of correlating molecular weight with BTB permeability in the 231-Br, as well as other intracardiac models [6]. Nonetheless, because PS externalization occurred in almost every tumor blood vessel but not normal vessels, positive immunostaining with the PS-targeting antibody successfully delineated individual brain metastases, even micrometastases, from normal brain tissues (Figure 2 and Figure 3).

We further demonstrated the development of PS-targeted imaging probes for sensitive and specific imaging of brain metastases. In addition to the 231-Br model, we also injected murine mammary cancer 4T1 cells intracardiacally into the immunocompetent mice. In contrast to multifocal lesions in the 231-Br mouse brain, fewer lesions, often a solitary lesion, were seen in the 4T1 mouse brain (Figure 5). The 4T1 brain metastases were larger and enhanced on T1-w post-contrast MRI (Figure 5b). Despite most of the 800CW-1N11 probes that were detected on tumor vascular ECs, extravascular signals of 800CW-1N11 were observed (Figure 5f), indicating that the BTB disruption is more severe in the 4T1 than the 231-Br brain metastases. Unlike many other cancer biomarkers, which are commonly confirmed with immunohistochemical or biological analyses of tissue specimens from the biopsy or resection of a tumor, there is a concern that PS may become externalized during the tissue resection and preparation process, and the tissues are no longer suitable for PS detection. Pre-injection of PS antibody to the patient will be necessary for accurate assessment. Alternatively, the development of an optical imaging probe targeting PS, as shown in this study, may be useful for the pathological diagnosis of occult metastases. Labeling with an NIR fluorescence dye enables the light to penetrate deeper because it has lower tissue absorption and scattering of light and is distinct from autofluorescence. Without the use of a secondary antibody for staining, the resected tissue chucks may be screened for the likelihood of brain metastases based on the observation of the optical signals (Figure 5a). Subsequent tissue sectioning will likely provide a definitive pathological diagnosis in conjunction with routine histology and immunohistochemistry (Figure 5b). Moreover, NIR optical imaging has been increasingly applied for the intraoperative detection of residual tumors and cancer metastases. The PS-targeting optical probe developed in this study may also be useful for neurosurgeons to identify occult metastases and tumor margins during the surgical resection of brain metastasis.

Despite its excellent sensitivity, the utility of optical imaging is largely limited by poor light penetration. We have previously attached an MRI contrast agent, iron oxide, to the PS antibody and demonstrated its feasibility for imaging tumor vasculature in a breast cancer mouse model [31]. MRI is known for its excellent spatial resolution. However, high concentrations of imaging probes are often required, thereby lowering the imaging sensitivity. In the present study, we have also labeled 1N11 F(ab’)_2_ with I-125 and successfully applied it for the autoradiography imaging of brain metastases, which provided a proof-of-concept for its potential for the in vivo nuclear imaging of brain metastases. Indeed, we have previously radiolabeled a PS antibody with ^74^As, which is a long-lived positron emitter, and used it for the PET imaging of prostate tumors in rats [32]. Several other radioisotopes, such as ^64^Cu for PET imaging and ^111^In for SPECT, have also been shown to image PS expression in various cancer models [33]. Unlike metabolic radiotracers, such as ^18^FDG for PET imaging, which often give high signals in the normal brain due to a high metabolic rate of neurons, the PS-radioisotope probe is more likely specific to brain metastases because PS is exposed only in tumor blood vessels. Thus, it will be interesting to explore the utility of the PS-targeted imaging probes for the PET/SPECT imaging of occult brain metastases.

The anti-cancer therapeutic effects of PS-targeting antibody have previously been studied. The antibodies bind to PS-exposed tumor vascular ECs and mediate the antibody-dependent cell-mediated cytotoxicity (ADCC) by monocytes and macrophages, resulting in tumor blood vessel collapse and consequently, tumor cell starvation and death [34]. The level of PS exposure was found to increase after chemotherapy or radiotherapy; thus, a combination with anti-PS antibodies enhanced treatment efficacy [34,35,36]. Given that WBRT is the standard-of-care treatment for brain metastases in clinics, it will be interesting to investigate the combination of WBRT with anti-PS antibodies in brain metastases. PS is also known for its contribution to anti-inflammatory and immunosuppressed responses through PS-PS receptor signaling [37,38]. By blocking the interaction, anti-PS antibodies were shown to successfully reprogram tumor-associated macrophages (TAMs) and myeloid-derived suppressor cells (MDSCs) and reactivate immunosuppressive effector T cells, which elicited anti-cancer innate and adaptive immunity [37,39].

Other strategies may also be considered to exploit PS-targeting antibodies for brain metastases treatments. Alternative to the diagnostic radioisotopes, radiotherapeutic isotopes, such as the β-emitters Yttrium-90 (^90^Y) or Lutetium-177 (^177^Lu), can be conjugated with PS-targeting antibodies. The advantages of β-emitters include their relatively high energy and shorter emission range. Yttrium-90 has a mean path length of ~3 mm, while ^177^Lu has an even shorter range and lower energy. After binding to PS on the blood vessels of brain metastases, the radiotherapeutics emit high energy to kill tumor vascular ECs and surrounding tumor cells. Importantly, the shorter traveling distance of β-emitters is less likely to cause unwanted damage to the normal brain. The lutetium-177-labeled prostate-specific membrane antigen (PSMA) has recently been approved by the FDA for treating metastatic prostate cancer patients. Thus, we foresee that the β-emitters labeled with PS may be specifically useful for targeting the clinically occult and micrometastases. Another antibody-based therapeutic strategy is to conjugate the antibody with a chemo-drug. Antibody–drug conjugates (ADCs) have recently shown clinical success in treating several types of cancer. A recent preclinical study reported the efficacy of anti-HER2 antibody-tubulysin ADC in preventing the brain metastases of HER2+ breast cancer in mouse models [40]. We consider anti-PS antibody a potential candidate for an ADC. Moreover, nanoparticle (NP) delivery systems may have advantages, including high drug payload and increased binding affinity due to multiple targeting molecules assembled on the surface of NPs [41,42]. We have recently developed various PS-targeted nano-delivery systems (PS-NPs) by functionalizing the nanoparticles with a PS-targeting antibody. While examining different types of PS-NPs, we observed an interesting finding that after binding to the PS-exposed cells, the lipid-based nanoparticles (LNPs)—but not other types of NPs, e.g., iron oxide—became internalized into the targeted cells [31,43]. It is believed that binding to the cell-surface PS leads to close apposition and subsequent fusion between the lipid layers of LNPs and the cell membrane. Thus, we anticipate that PS-targeting LNP may serve as an effective carrier to deliver anti-cancer drugs to brain metastases by bypassing the BTB through receptor-mediated transcytosis.

Because PS is the same molecule and has the same distribution and regulation in all mammalian species, it is plausible that the mouse data will extrapolate to humans. However, the level of PS exposure in brain metastases may vary between the mouse models and patients. Future clinical trials are warranted to evaluate PS exposure in brain metastasis patients with PS antibodies or PS-targeted imaging probes. While the exact mechanisms underpinning the high-level expression of PS on blood vessels of brain metastases are unclear, the factors related to oxidative stress and inflammatory cytokines, such as the TNF-α overexpression observed in this study (Figure 4), are likely playing an important role in flipping PS to the cell surface.

## 5. Conclusions

In summary, we demonstrate a novel strategy for targeting brain metastases based on our findings that extensive PS is exposed on the vascular endothelia of brain metastases but not the normal brain. The high sensitivity and specificity of PS-targeting antibody enables individual metastases, even micrometastases containing an intact BTB, to be clearly delineated. Given its intravascular localization and the lack of a need to cross the BTB, PS appears to be a useful biomarker for the development of imaging and targeted therapeutics for brain metastases.

## Figures and Tables

**Figure 1 cancers-16-03088-f001:**
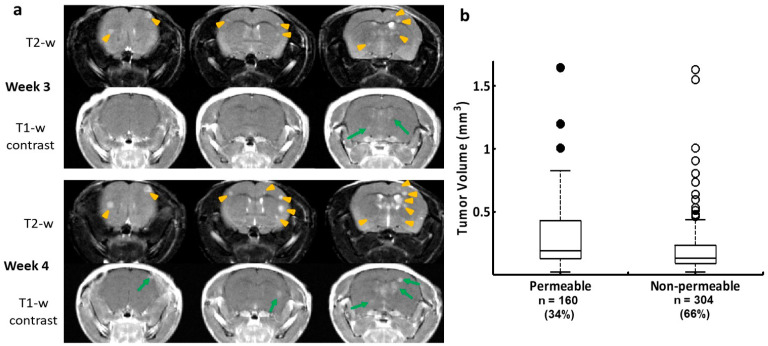
Longitudinal MRI monitoring of the intracranial distribution of brain metastases and permeability of the BTB in the 231-Br model. (**a**) Three consecutive coronal sections of high-resolution MRI were obtained from a representative mouse brain bearing 231-Br metastases. At week 3, T_2_-w images revealed multiple hyperintense metastases across the brain (yellow arrowheads), only two of which were enhanced on T_1_-w post-contrast images (green arrows), indicating a leaky BTB. At week 4, several new lesions appeared on T_2_-w images (yellow arrowheads), while those lesions seen on the prior scan became larger. An increased number of the contrast-enhanced lesions were seen at this time (green arrows). (**b**) A total of 464 231-Br brain metastases were identified by MRI, including non-permeable (*n* = 304) and permeable (*n* = 160) metastases based on T_1_-w post-contrast images. A plot of permeability versus size indicated that larger metastases tend to be leaky. However, there was no significant difference in tumor size between the permeable and non-permeable metastases (*p* = 0.1).

**Figure 2 cancers-16-03088-f002:**
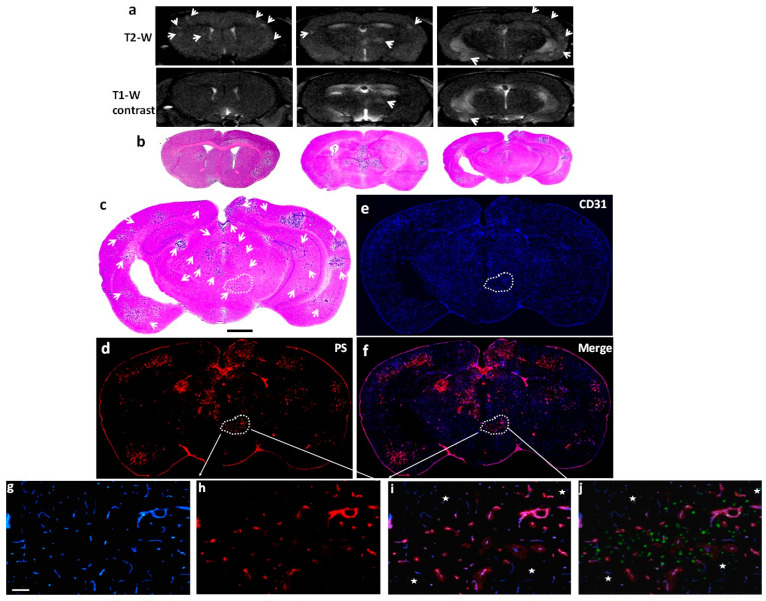
PS exposed exclusively in vascular endothelial cells of brain metastases. (**a**) Multiple brain lesions were detected on three consecutive coronal sections of T_2_-w images (arrow) in a 231-Br mouse, while a few lesions were enhanced on T_1_-w post-contrast images (arrow). Immediately after MRI, the mouse was injected i.v. with 1N11 (150 µg). Four hours later, the mouse was perfused, and the brain was dissected. (**b**) Corresponding H&E sections depicted more microscopic lesions that were invisible by MRI. (**c**–**f**) Diffuse brain metastases were labeled on one of the H&E sections (arrow; (**c**)). Immunofluorescence staining on a consecutive section showed that 1N11 (red; (**d**)) localized to every tumor lesion, even microscopic lesions identified by H&E staining (arrow; bar = 1 mm; (**c**)). The binding of 1N11 to tumor vessels (CD31, blue; (**e**)) was confirmed by the magenta color in the merged images (**f**). (**g**–**j**) A region containing positive 1N11 (outlined in (**d**)) was selected and magnified (bar = 100 µm). The merged image showed that 1N11 (red, (**h**)) co-localized with almost every CD31-positive tumor vessel (blue, (**g**)) to give a magenta color (**i**). Vessels in nearby normal brain (star, (**i**)) were not stained by 1N11. The tumor regions were distinguished from normal brain by the presence of GFP in the tumor (**j**).

**Figure 3 cancers-16-03088-f003:**
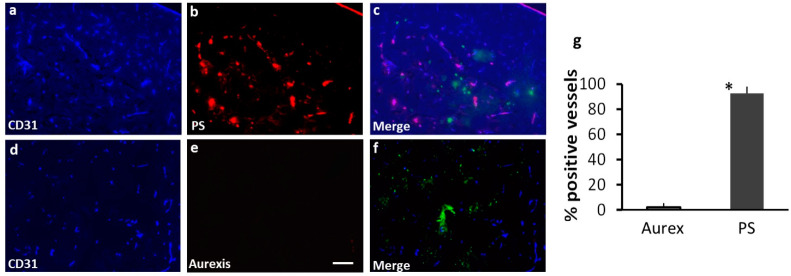
Staining with 1N11 was antigen-specific. 231-Br mice were injected i.v. with either 1N11 or the control antibody, Aurexis (150 µg). Four hours later, the mice were perfused, and the brains were dissected. (**a**–**c**) A brain region containing a tumor lesion was stained with anti-CD31 for blood vessels (**a**) and co-stained for 1N11 (**b**). The merged image showed that 1N11 co-localized with CD31-positive tumor vessels. Vessels in nearby normal brain (containing no GFP) were not stained by 1N11 (**c**). (**d**–**f**) In contrast, the control antibody, Aurexis, showed no staining of blood vessels of brain metastases (bar = 100 µm). (**g**) For the group of 231-Br mice, the percentage of PS-positive vessels in brain metastases was 93 ± 5%, while it was only 2 ± 2% for the control antibody, Aurexis. * *p* < 0.01.

**Figure 4 cancers-16-03088-f004:**
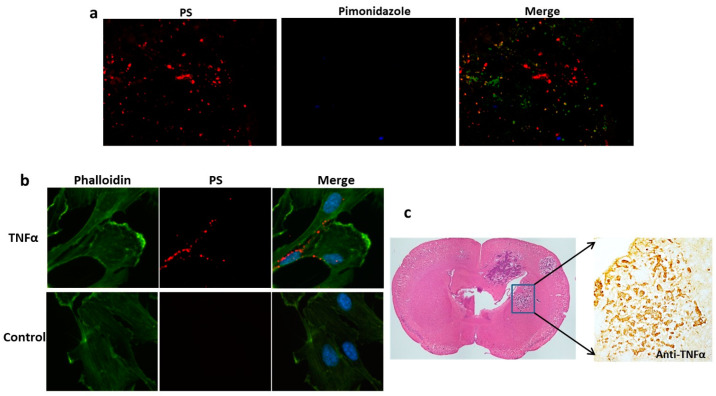
TNF-α-induced PS externalization in vascular endothelial cells and the overexpression of TNF-α was detected in brain metastases. (**a**) A group of 231-Br mice receiving i.v. 1N11 followed by i.v. pimonidazole was used to study tumor hypoxia. A representative region containing positive 1N11 staining (red) was co-stained for pimonidazole (blue). There was essentially no positive pimonidazole staining in the brain metastasis (GFP). (**b**) HUVECs were treated with/without TNF-α (20 ng/mL) for 24 h before fixation, then immunocytostained with 1N11 (red), cytoskeleton (phalloidin, green), and nuclear (DAPI, blue). The merged image showed numerous cell surface-exposed PS in the TNF-α-treated cells, while there was no PS exposure in the control cells. (**c**) Anti-TNF-α staining revealed marked expression of TNF-α in the 231-Br brain metastasis but not in normal brains.

**Figure 5 cancers-16-03088-f005:**
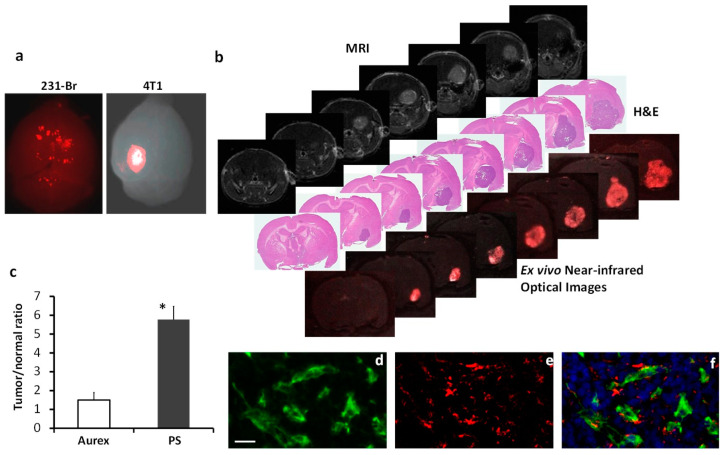
PS-targeted NIR imaging of brain metastases. NIR dye, IRDye-800CW-labeled 1N11 F(ab’)_2_, was injected into the mice bearing 231-Br (4 wks post-tumor cell injection) or 4T1 (2 wks post-tumor cell injection) brain metastases. (**a**) Twenty-four hours later, ex vivo NIR imaging of the whole brain detected distinct 800CW signals from multiple regions of a 231-Br brain, while a clear contrast of a single brain lesion was seen in a 4T1 brain. (**b**) A series of unstained coronal brain cryosections (10 µm, bottom row) was obtained after the brain in (**a**) was imaged. The NIR imaging revealed a clear tumor contrast on the series of brain sections, which correlated well with H&E staining (middle row) of tumors and T1-contrast-enhanced MR images (top row). (**c**) Quantification of the light intensity in the tumor versus the contralateral normal brain obtained a ratio of 5.7 ± 0.8 with 800CW-1N11, which was significantly higher than that of the control antibody, 800CW-Aurexis (1.5 ± 0.4; * *p* < 0.05). (**d**) Tumor vascular endothelial cells were immunostained with anti-CD31 (green) in the cryosections adjacent to those used in (**b**). 800CW-1N11 signals (red) from the same field were detected with an NIR filter set (**e**). (**f**) The merged image revealed extensive 800CW-1N11 bound to vascular endothelial cells, while some were detected in the extravascular space (DAPI, blue).

**Figure 6 cancers-16-03088-f006:**
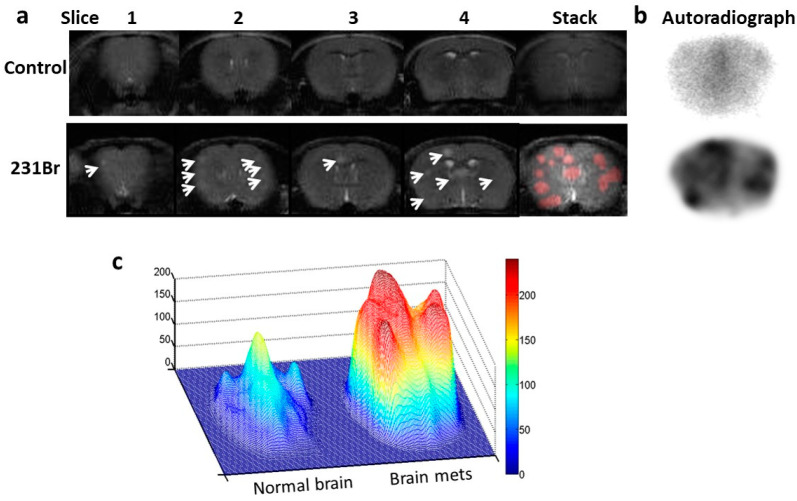
Autoradiography imaging of I-125-labeled 1N11 in targeting brain metastases. (**a**) T_2_-w MR images of 4 consecutive 1 mm thick coronal slices covering 4 mm brain tissues post-bregma were acquired from a normal control mouse brain (**top**) and a 231-Br brain (**bottom**), respectively. Multiple tumors in the 231-Br brain were identified (arrow). An image stack across the 4 slices was created for each animal. For the 231-Br brain, the tumor lesions (red) from each slice were projected on the stacked image to correlate with the autoradiography study. (**b**) After the MRI, each mouse was given i.v. 60 µCi ^125^I-1N11. Forty-eight hours later, the mice were perfused, and mouse brains were dissected. Using a mouse brain matrix, 4 mm thick brain tissues post-bregma, correlating with the MRI, were cut from each mouse brain and laid with the cutting face on the autoradiograph film. After 12 h of incubation, autoradiograph images showed multiple hot spots on the tumor brain, while a clean background signal was observed on the normal brain. There was a general spatial correlation between tumor lesions on the MRI and hot spots on the autoradiograph. (**c**) Significantly higher uptake of ^125^I-1N11 was observed in individual brain metastases as compared to normal brain tissues (a ratio of 3.6 ± 0.8; *p* < 0.05).

## Data Availability

All data are available upon request from the corresponding author.

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
