# Peer review of "Exposed Phosphatidylserine as a Biomarker for Clear Identification of Breast Cancer Brain Metastases in Mouse Models"

_cancers, 2024, doi:10.3390/cancers16173088_

Round 1

Reviewer 1 Report

Comments and Suggestions for Authors

The manuscript Targeting Exposed Phosphatidylserine Enables the Clear Demarcation of Brain Metastases in Mouse Models” described the approaches to applying anti-PS antibodies for the systemic targeting and imaging of brain metastasis in mice models. Since brain metastasis is intractable in clinical settings, this method is of significant interest to the field. The manuscript is well-structured, and the experimental procedure is stringently described. In the discussion section, the potential and the foresight of the application for such a targeting method are also well demonstrated. I recommend the manuscript to be published after the issue below is well addressed:

1.      Systemic administration of 4T1 is usually extremely aggressive in the Balb/c model most mice could not endure the multi-organ metastasis burden within 2 weeks according to most literatures. The reviewer is curious about the 2-5 weeks imaging timeline of the mice metastasis model and would recommend that the author specify the status of mice such as behavior and body weight and mark the timeline clearly in the figure of each study to avoid any confusion.

2.      There is a confusing typo in the abstract: “brain-tumor-barrier (BTB)”

Author Response

  1. Systemic administration of 4T1 is usually extremely aggressive in the Balb/c model most mice could not endure the multi-organ metastasis burden within 2 weeks according to most literatures. The reviewer is curious about the 2-5 weeks imaging timeline of the mice metastasis model and would recommend that the author specify the status of mice such as behavior and body weight and mark the timeline clearly in the figure of each study to avoid any confusion.

We agree that 4T1 cells are aggressive, if injected intravenously, most of the cells will be stuck in lungs to form lung metastases. In this study, we injected the cells intracardiacally and started to image the mouse 2 weeks post injection. For the 231-Br mice, the imaging timeline was up to 5 weeks. For the 4T1, brain tumor was typically detected at week 2 when the animals’ conditions were not deteriorating significantly. We have clarified the imaging time in the figure.

  1. There is a confusing typo in the abstract: “brain-tumor-barrier (BTB)”

Corrected.

Reviewer 2 Report

Comments and Suggestions for Authors

The manuscript presents an innovative approach to targeting brain metastases by utilizing exposed phosphatidylserine (PS) as a biomarker, which is a significant contribution to the field.

However, several aspects of the paper require revision:

-The manuscript would benefit from clearer organization, particularly in the separation of Results from the Discussion. Ensuring each section serves its intended purpose will greatly improve readability and help the audience distinguish between experimental findings and their interpretation. Some data currently discussed in the Discussion should be moved to the Results section.

-The current title, "Targeting Exposed Phosphatidylserine Enables the Clear Demarcation of Brain Metastases in Mouse Models," is descriptive but could be revised to emphasize the translational potential of this work:

"Phosphatidylserine Targeting Enables Precise Detection of Brain Metastases in Mouse Models" OR "Exposed Phosphatidylserine as a Biomarker for Clear Identification of Brain Metastases in Mouse Models"

-While the introduction addresses the challenges of brain metastasis treatment and diagnosis, it could benefit from a clearer articulation of the specific knowledge gaps that the study aims to fill. The problem of poor prognosis and the limitations of current therapies are discussed, but the introduction should more explicitly state how targeting PS is a novel or superior approach compared to existing strategies. Expanding on why PS targeting could overcome the challenges of brain metastasis treatment would provide a stronger rationale for the study.

- The introduction explains that PS is externalized in non-apoptotic endothelial cells in tumor vasculature, but it lacks sufficient detail on why this happens and how this phenomenon can be exploited for brain metastasis targeting. Providing more mechanistic insights into how oxidative stress and the tumor microenvironment lead to PS exposure would enhance the scientific foundation of the study. A more thorough explanation of the reversible nature of PS externalization and its significance in this context is also needed.

- While PS is exposed on tumor endothelial cells, it would be beneficial to discuss any potential off-target effects or the extent of PS exposure in non-tumor tissues, which could affect the safety and efficacy of the proposed targeting approach.

- The authors should more directly address how targeting PS in mouse models could be applied in the clinic and what hurdles must be overcome before translation to human patients.

- The PS-targeting approach has been explored in other cancer models, so the authors should elaborate on how their findings significantly advance the current understanding or provide new avenues for brain metastasis detection and treatment.

- The manuscript implies that PS targeting bypasses BTB, but the mechanisms by which this occurs need more explanation. How does PS exposure relate to the permeability of the BTB in different stages of metastasis? Does the antibody actually permeate the BTB, or is PS exposure indicative of compromised BTB integrity?

- Although the study focuses on imaging brain metastases, the authors briefly mention the therapeutic potential of PS targeting. This aspect should be expanded. How could PS-targeting be integrated with therapeutic agents? Can this strategy be used for drug delivery or combined with existing treatments like radiation or chemotherapy?

-Two different MRI systems (9.4T Varian and 7T Bruker) were used. It would be helpful to clarify whether these two systems were used interchangeably or in different sets of animals, and if so, how differences in image acquisition and resolution were accounted for.

- While the use of 4T1 and 231-Br cells is appropriate for brain metastasis modeling, it would be helpful to provide a brief justification for using these specific models. This could include a note on their relevance to human breast cancer metastasis in terms of biological behavior or clinical relevance.

-The rationale for choosing the 4-hour timepoint post-antibody injection is not explained. Why was this timepoint chosen, and could alternative timepoints provide different results?

-While spatial correlation between autoradiography and MRI is mentioned, there is no detail on how this was accomplished (e.g., software or manual overlay).

-The use of a one-tailed, unpaired Student’s t-test is mentioned, but it may not be appropriate for all comparisons. A two-tailed test might be more suitable unless there is a clear rationale for using a one-tailed test. A discussion of statistical considerations, including the appropriateness of tests and corrections for multiple comparisons, is essential to ensure accurate interpretation of the data.

- While it's plausible that PS expression will translate to humans, it needs a stronger emphasis on the limitations of this assumption. The sentence "Because PS is the same molecule and has the same distribution and regulation in all mammalian species..." is too broad and requires a more cautious stance, acknowledging interspecies differences and the complexity of translating preclinical findings to human clinical outcomes.

-Radiotherapeutics involving PS-targeting antibodies sound promising, but challenges such as delivery to the brain and toxicity concerns should be discussed more deeply. Moreover, the potential immunogenicity of using murine-derived antibodies like 1N11 in humans should be addressed.

-The discussion touches on various imaging techniques (NIR, PET/SPECT, MRI), but it does not sufficiently explain how the proposed PS-targeting strategy compares in terms of sensitivity, depth of penetration, or ease of clinical use. Clarifying the advantages or unique contributions of PS-targeted imaging relative to established modalities will help readers understand the added value of this work.

Comments on the Quality of English Language

MINOR EDITING

Author Response

The manuscript presents an innovative approach to targeting brain metastases by utilizing exposed phosphatidylserine (PS) as a biomarker, which is a significant contribution to the field.

Thank you for your positive review and constructive comments. We have responded the reviewer's comments as follows and incorporate the changes into the manuscript.

However, several aspects of the paper require revision:

-1. The manuscript would benefit from clearer organization, particularly in the separation of Results from the Discussion. Ensuring each section serves its intended purpose will greatly improve readability and help the audience distinguish between experimental findings and their interpretation. Some data currently discussed in the Discussion should be moved to the Results section.

We have revised the manuscript accordingly.

-2. The current title, "Targeting Exposed Phosphatidylserine Enables the Clear Demarcation of Brain Metastases in Mouse Models," is descriptive but could be revised to emphasize the translational potential of this work:

"Phosphatidylserine Targeting Enables Precise Detection of Brain Metastases in Mouse Models" OR "Exposed Phosphatidylserine as a Biomarker for Clear Identification of Brain Metastases in Mouse Models"

The title is changed to ‘Exposed Phosphatidylserine as a Biomarker for Clear Identification of Breast Cancer Brain Metastases in Mouse Models’.

  1. While the introduction addresses the challenges of brain metastasis treatment and diagnosis, it could benefit from a clearer articulation of the specific knowledge gaps that the study aims to fill. The problem of poor prognosis and the limitations of current therapies are discussed, but the introduction should more explicitly state how targeting PS is a novel or superior approach compared to existing strategies. Expanding on why PS targeting could overcome the challenges of brain metastasis treatment would provide a stronger rationale for the study.

As alluded in the Introduction, unlike visceral cancers, in which tumor capillaries are leaky due to extensive angiogenesis, the clinically occult or micro-metastases are impermeable to diagnostic or therapeutic agents. The vascular luminal surface-exposed PS is directly accessible to systemically administered diagnostic or therapeutic agents. Also distinct from the strategy targeting physiological receptors such as transferrin, insulin and LDL receptor, which express in not only tumor blood vessels but also normal vascular endothelial cells, PS exposure is specific to tumor vasculature.  Thus, we consider PS targeting a novel approach that is superior to the existing approaches. We have expanded the Introduction to state the novelty of targeting PS for brain metastases and its potential for overcoming the challenges of brain metastasis treatment.      

-4. The introduction explains that PS is externalized in non-apoptotic endothelial cells in tumor vasculature, but it lacks sufficient detail on why this happens and how this phenomenon can be exploited for brain metastasis targeting. Providing more mechanistic insights into how oxidative stress and the tumor microenvironment lead to PS exposure would enhance the scientific foundation of the study. A more thorough explanation of the reversible nature of PS externalization and its significance in this context is also needed.

More details on plausible mechanisms underpinning PS exposure are now provided in the Introduction.

-5.  While PS is exposed on tumor endothelial cells, it would be beneficial to discuss any potential off-target effects or the extent of PS exposure in non-tumor tissues, which could affect the safety and efficacy of the proposed targeting approach.

In addition to tumor endothelial cells, PS is found to expose on tumor cells. Thus, PS-targeted therapeutics has the potential of kill both tumor vascular ECs and tumor cells. Moreover, PS has been reported to express on some immune cells in the TME, which suppresses anti-cancer immune responses. Thus, blocking PS signaling with PS targeting antibody was found to remodel the tumor immune environment and enhance anti-cancer immunotherapy. However, previous studies did not detect PS exposure in normal tissues, even the highly angiogenic endometrial tissues (Ran, Int. J. Radiation Oncology Biol. Phys. 2002), lowering a low likelihood of PS-targeting antibodies to cause potential off-target effects.

-6. The authors should more directly address how targeting PS in mouse models could be applied in the clinic and what hurdles must be overcome before translation to human patients.

We have included brief discussion about the translational potential of targeting PS and the possible discrepancy in findings between animal models and humans in the Discussion section.

-7. The PS-targeting approach has been explored in other cancer models, so the authors should elaborate on how their findings significantly advance the current understanding or provide new avenues for brain metastasis detection and treatment.

As in our previous response to the reviewer, unlike visceral cancers, in which tumor capillaries are leaky due to extensive angiogenesis, the clinically occult or micro-metastases are impermeable to diagnostic or therapeutic agents. The vascular luminal surface-exposed PS is directly accessible to systemically administered diagnostic or therapeutic agents. Thus, we consider that PS targeting is a novel approach that may be utilized for brain metastasis diagnosis and treatment.

-8. The manuscript implies that PS targeting bypasses BTB, but the mechanisms by which this occurs need more explanation. How does PS exposure relate to the permeability of the BTB in different stages of metastasis? Does the antibody actually permeate the BTB, or is PS exposure indicative of compromised BTB integrity?

As shown in the data of Figure 1, although larger 231-Br metastases tended to be more leaky, there was no significant correlation between tumor size and vascular permeability. We conducted pharmacological studies of PS antibodies at the late stage of the 231-Br metastasis (4-5 wks post injection). PS staining, as shown in Figs. 2 and 3, detected the PS antibodies co-localized with CD31+  tumor vascular ECs. There was essentially no extravascular positive staining, indicating the BTB in the 231-Br was impermeable to PS antibodies. However, in the 4T1 model (Fig. 5), brain metastases were much larger than the 231 mets. In addition to the signals co-localizing with tumor ECs, we did visualize PS antibodies in extravascular space. However, as the Reviewer 1 pointed out, the short life span of the 4T1 mice did not allow us to conduct longitudinal studies.   

-9. Although the study focuses on imaging brain metastases, the authors briefly mention the therapeutic potential of PS targeting. This aspect should be expanded. How could PS-targeting be integrated with therapeutic agents? Can this strategy be used for drug delivery or combined with existing treatments like radiation or chemotherapy?

We have discussed about the potential of combining PS targeting antibody with standard of care treatment such as chemotherapy and radiation, and anti-cancer immunotherapy such as anti-PD-1/PD-L1, and the development of PS-targeted novel therapeutics such as radiotherapeutics and antibody-drug conjugates (ADCs). We have expanded our discussion on the potential of developing PS-targeted nanocarriers for drug delivery to brain tumors.   

-10. Two different MRI systems (9.4T Varian and 7T Bruker) were used. It would be helpful to clarify whether these two systems were used interchangeably or in different sets of animals, and if so, how differences in image acquisition and resolution were accounted for.

As indicated in the manuscript, the current study was conducted in two institutions, where the 9.4T Varian is available at UTSW, while the 7T Bruker is at Wake Forest. Despite the different systems, we ensured that the acquisition and spatial resolution (FOV 20 x 20 mm; matrix 256 x 256) were commensurate.

-11.  While the use of 4T1 and 231-Br cells is appropriate for brain metastasis modeling, it would be helpful to provide a brief justification for using these specific models. This could include a note on their relevance to human breast cancer metastasis in terms of biological behavior or clinical relevance.

The triple negative breast cancer is a highly aggressive breast cancer subtype that tends to metastasize to the brain. Both 4T1 and 231-Br cells were derived from the triple negative breast cancer. The 231-Br model was established in the immunocompromised nude mice, while the immunocompetent balb/c mice were used to create the 4T-1 breast cancer brain metastasis model       

-12. The rationale for choosing the 4-hour timepoint post-antibody injection is not explained. Why was this timepoint chosen, and could alternative timepoints provide different results?

We consider that the 4-hour circulation allows sufficient time for the antibodies to bind PS. Previous pharmacological studies of such PS antibodies by us and others have used either 2-hour or 4-hour times in different tumor models. We do not anticipate different results from alternative timepoints because the BTB of brain metastases in mice is mostly impermeable to the macromolecular antibodies.

-13. While spatial correlation between autoradiography and MRI is mentioned, there is no detail on how this was accomplished (e.g., software or manual overlay).

As in a new video provided as the supplemental data, we manually overlaid the two images. Although the two images varied in spatial resolution, visual comparison showed a good spatial correlation between the two imaging modalities.

-14. The use of a one-tailed, unpaired Student’s t-test is mentioned, but it may not be appropriate for all comparisons. A two-tailed test might be more suitable unless there is a clear rationale for using a one-tailed test. A discussion of statistical considerations, including the appropriateness of tests and corrections for multiple comparisons, is essential to ensure accurate interpretation of the data.

Agree. We re-ran the statistical analyses with a two-tailed test, which gave similar results. We corrected it in the Methods

-15. While it's plausible that PS expression will translate to humans, it needs a stronger emphasis on the limitations of this assumption. The sentence "Because PS is the same molecule and has the same distribution and regulation in all mammalian species..." is too broad and requires a more cautious stance, acknowledging interspecies differences and the complexity of translating preclinical findings to human clinical outcomes.

Agree. We have rephrased the statement. ‘However, the levels of PS exposure in brain metastases may vary between the mouse models and patients. Future clinical trials are warranted to evaluate PS exposure in brain metastasis patients with PS antibodies or the PS-targeted imaging probes.   

-16. Radiotherapeutics involving PS-targeting antibodies sound promising, but challenges such as delivery to the brain and toxicity concerns should be discussed more deeply. Moreover, the potential immunogenicity of using murine-derived antibodies like 1N11 in humans should be addressed.

Agree. For any type of therapeutics, risk versus benefit should always be carefully considered. Given the PS exposure specific to blood vessels of brain metastases, PS-targeted radioisotopes are less likely deposited in normal brain. However, it is plausible that the tumor-bound PS radiotherapeutics emit radiation that travels to normal brain and causes off-target side effects. An advantage of β- emitters is their shorter emission path lengths. Yttrium-90 (90Y) has a mean path length of ~3 mm, while Lutetium-177 (177Lu) has even shorter range and lower energy. 177Lu labeled prostate specific membrane antigen (PSMA) has recently been approved by FDA for treating metastatic prostate cancer patients. Thus, we foresee that the β- emitters labeled PS may be specifically useful for targeting the clinically occult/micrometastases. In terms of potential immunogenicity, 1N11 is a fully human mAb. Bavituximab is a chimeric PS-targeting antibody that is currently in clinical testing.

-17. The discussion touches on various imaging techniques (NIR, PET/SPECT, MRI), but it does not sufficiently explain how the proposed PS-targeting strategy compares in terms of sensitivity, depth of penetration, or ease of clinical use. Clarifying the advantages or unique contributions of PS-targeted imaging relative to established modalities will help readers understand the added value of this work.

We have expanded the discussion about the utility and limitation of various imaging modalities potentially for imaging PS exposure.

Reviewer 3 Report

Comments and Suggestions for Authors

In this paper titled ‘Targeting Exposed Phosphatidylserine Enables the Clear Demarcation of Brain Metastases in Mouse Models’, Wang et al., report that Phosphatidylserine (PS) is abundantly externalized on the surface of viable endothelial cells in tumor blood vessels of breast cancer brain metastases. Using a PS-targeting antibody 1N11 conjugated with a fluorescence dye or a radioisotope, the authors were able to visualize and clearly delineate the individual brain metastases by optical imaging or autoradiography. The authors demonstrated a novel strategy of targeting breast cancer brain metastases for imaging or targeted therapies, based on the nature of PS exposure. Overall, the study is novel and scientifically sound. I therefore recommend acceptance of the manuscript with the following minor suggestions.

1)    In this study, the authors tested their hypothesis using only breast cancer models. I suggest that the authors mention ‘breast cancer’ in the title and abstract, as we don’t know if this applies to other cancer types.

2)    Typos in Fig. 3 and Fig. 4 legend, ‘231-Br’ instead of ‘213-Br’.

Author Response

1)    In this study, the authors tested their hypothesis using only breast cancer models. I suggest that the authors mention ‘breast cancer’ in the title and abstract, as we don’t know if this applies to other cancer types.

We have made the change, as suggested.

2)    Typos in Fig. 3 and Fig. 4 legend, ‘231-Br’ instead of ‘213-Br’.

Corrected.

Round 2

Reviewer 2 Report

Comments and Suggestions for Authors

The authors addressed the issues I raised.

Comments on the Quality of English Language

minor editing, for example line 83 unknow